# Plant and Soil Microbial Diversity Co-Regulate Ecosystem Multifunctionality during Desertification in a Temperate Grassland

**DOI:** 10.3390/plants12213743

**Published:** 2023-10-31

**Authors:** Yeming Zhang, Xiuli Gao, Ye Yuan, Lei Hou, Zhenhua Dang, Linna Ma

**Affiliations:** 1Ministry of Education Key Laboratory of Ecology and Resource Use of the Mongolian Plateau & Inner Mongolian Key Laboratory of Grassland Ecology, School of Ecology and Environment, Inner Mongolia University, Hohhot 010021, China; zyeming98@163.com; 2State Key Laboratory of Vegetation and Environmental Change, Institute of Botany, The Chinese Academy of Sciences, Beijing 100093, China; 3Chinese Research Academy of Environmental Sciences, Beijing 100012, China; gaoxiuli1551070@163.com; 4School of Urban Planning and Design, Shenzhen Graduate School, Peking University, Shenzhen 518055, China; yuanye0105@163.com; 5Beijing Municipal Pollution Source Management Center, Beijing 100089, China; hl2641908@126.com

**Keywords:** desertification, grassland, plant diversity, microbial diversity, multifunctionality

## Abstract

Biodiversity plays a crucial role in driving multiple ecosystem functions in temperate grasslands. However, our understanding of how biodiversity regulates the impacts of desertification processes on ecosystem multifunctionality (EMF) remains limited. In this study, we investigate plant diversity, soil microbial diversity (fungal, bacterial, archaeal, and arbuscular mycorrhizal fungal (AMF) diversity), soil properties (soil water content, pH, and soil clay content), and multiple ecosystem functions (soil N mineralization, soil phosphatase activity, AMF infection rate, microbial biomass, plant biomass, and soil C and nutrients (N, P, K, Ca, Fe, Na, Cu, Mg, and Mn)) at six different grassland desertification intensities. The random forest model was conducted to assess the importance of soil properties, plant diversity, and soil microbial diversity in driving EMF. Furthermore, a structural equation model (SEM) was employed to analyze the indirect and direct impacts of these predictors on EMF. Our study showed that plant, soil bacterial, fungal, and archaeal diversity gradually decreased with increasing desertification intensity. However, only AMF diversity was found to be less sensitive to desertification. Similarly, EMF also showed a significant decline with increasing desertification. Importantly, both plant and soil microbial diversity were positively associated with EMF during desertification processes. The random forest model and SEM revealed that both plant and soil microbial diversity were identified as important and direct predictors of EMF during desertification processes. This highlights the primary influence of above- and below-ground biodiversity in co-regulating the response of EMF to grassland desertification. These findings have important implications for planned ecosystem restoration and sustainable grassland management.

## 1. Introduction

Biodiversity plays a fundamental role in the maintenance of ecosystem multifunctionality (simultaneously providing multiple ecosystem functions, EMF) in terrestrial ecosystems [1,2,3,4]. These functions predominantly include plant production, litter decomposition, C storage, nutrient cycles, and greenhouse emissions [5,6,7]. Recently, numerous studies have demonstrated the positive and precise impacts of plant and microbial diversity on EMF in natural ecosystems. Therefore, any biodiversity loss may negatively impact multiple ecosystem functions and significantly decrease EMF [8,9,10]. However, a comprehensive assessment of the roles of plant and microbial diversity in regulating the EMF of disturbed ecosystems remains largely uninvestigated.

Grassland is one of the most common vegetation types in the world, accounting for 46% of the global land area [11]. China has the third-largest grassland area in the world, accounting for approximately a quarter of the country’s land area [12]. Grassland provides various ecosystem functions and services, including C storage, biogeochemical cycles, biodiversity conservation, and food production for human well-being [13]. During recent decades, Chinese grasslands have experienced large-scale desertification (i.e., grassland degradation) as a result of human and climatic disturbances (e.g., overgrazing, farming, and drought), which affect the lives of over 47.9 million people [14,15,16]. Overgrazing leads to the loss of plant cover and a decline in soil quality due to the trampling of plants by livestock [17]. These variations in plant and soil properties may negatively impact biodiversity and subsequently influence ecosystem functions.

Plant diversity is a critical predictor for explaining the variations in multiple ecosystem functions under disturbed ecosystem paradigms. A recent study showed that grazing disturbance can negatively impact plant diversity, soil bacterial diversity, and EMF under different grazing intensities. Grazing reduces EMF primarily by decreasing plant diversity rather than bacterial diversity [2]. By contrast, soil microbial diversity, including bacterial, fungal, archaeal, and AMF diversity, may also act as a primary driver of EMF during desertification, as diverse soil microbes can perform critical ecosystem functions by decomposing litter, defining soil structure, regulating nutrient cycling, and supporting productivity [14]. To date, although most studies have demonstrated the effects of desertification on biodiversity and ecosystem functions, the importance of above- and below-ground biodiversity in mediating the impact of desertification intensities on EMF remains unclear [18].

A comprehensive understanding of the impacts of above- and below-ground diversity on EMF at different grassland desertification stages is crucial for developing practical solutions for combating global grassland desertification. In this study, we assessed plant richness, microbial richness (bacterial, fungal, archaeal, and AMF richness), and 15 ecosystem functions (i.e., soil-available N, phosphatase activity, AMF infection rate, phospholipid fatty acids, plant biomass, and soil nutrients (N, Ca, K, Fe, Na, Mg, C, Cu, Mn, and P)) at different desertification stages (potential desertification, light desertification, moderate desertification, heavy desertification, severe desertification, and very severe desertification) in a temperate grassland. We hypothesized that (a) plant and soil microbial diversity gradually decrease during grassland desertification processes [6]; (b) EMF also exhibits a gradual decrease trend with increasing desertification intensity based on the fact that desertification causes soil degradation and reduces potential productivity [2]; and (c) plant and soil microbial diversity co-regulate EMF during desertification processes in temperate grassland based on the fact that plant and microbial communities modulate the supply and conversion of C and nutrients in grasslands [19].

## 2. Materials and Methods

### 2.1. Study Site

This study was conducted at the Hulunber meadow steppe, located at Xiertala farm (49°19′ N, 120°02′ E, 630 m a.s.l.), Inner Mongolia, China. The mean annual air temperature ranged from −3 °C to −1 °C, and the mean annual precipitation was approximately 350 mm (1980–2010; http://data.cma.cn (30 January 2023)). According to FAO classification, the soil in this region is classified as chestnut soil. The natural vegetation is primarily composed of perennial grass species such as *Stipa baicalensis* and *Leymus chinensis*, as well as other plentiful plant species, including *Artemisia tanacetifolia*, *Artemisia frigida*, and *Serratula centauroides*. Throughout various stages of desertification, overall plant biomass ranged from 5 to 150 g m^−2^ (Appendix A).

### 2.2. Sampling and Processing

A space-for-time technique was utilized in this study to describe different stages of grassland desertification [20]. Potential desertification (PD), light desertification (LD), moderate desertification (MD), heavy desertification (HD), severe desertification (SD), and very severe desertification (VSD) were the six desertification intensities [21] (Appendix A). The PD stage was considered the natural grassland with high vegetation biomass, species richness, and soil quality (Appendix A).

A total of 90 soil samples were acquired, with 15 samples collected for each desertification stage (site). Field sampling was performed in mid-August 2022, during the peak growing season. The site locations were approximately 5 to 10 km apart. A 100-m transect was laid out at each site, and fifteen 1 × 1 m^2^ quadrats were randomly distributed along the transect, with a minimum interval of 15 m between quadrats. Three soil samples with a diameter of 5 cm were obtained at a depth of 15 cm within each quadrat. The soil samples were bulked, homogenized in the field, and kept in a 4 °C cooler for 7 days before being brought to the laboratory. The soil samples were sieved through a 2 mm sieve.

Air-dried subsamples for soil physical and chemical property tests were crushed into a fine powder. Subsamples for phosphatase activity and soil microbial diversity were maintained at −80 °C. The number of species identified in each quadrat was recorded as plant richness. Plant species diversity was estimated using plant richness. All plant tissue from each quadrat was harvested to determine the aboveground biomass of vascular plants. To quantify root biomass, a cylindrical root sampler was used to gather three soil cores in each quadrat. All plant tissue samples were dried in an oven at 65 °C until they reached a consistent weight.

### 2.3. Measurements of Soil Functions and Properties

The levels of soil total C and N were determined using a vario EL *III* CHNOS Elemental Analyzer (Elementar Analysensysteme GmbH, Langenselbold, Germany). The levels of soil P, Mn, Ca, Cu, Mg, K, Na and Fe were determined using an iCAP 6300 ICP-OES Spectrpmeter (Thermo Fisher, Waltham, MA, USA). Soil phosphatase activity was measured by the release of phenol from samples incubated with p-nitrophenyl phosphate (0.5%) at 37 °C [22]. Soil inorganic N was determined using a FIAstar 5000 analyzer (Foss Tecator, Hillerød, Denmark). The potential N mineralization rate was estimated as the difference between initial and final inorganic N levels before and after 14 days of incubation at 25 °C [20]. Arbuscular mycorrhizal fungal infection (AMF) rate was calculated by the number of root tips colonized by fungi divided by the total number of root segments examined [23].

### 2.4. Measurements of Microbial Biomass and Diversity

Phospholipid fatty acid (PLFA) analysis was used to assess the microbial biomass [24]. The PLFAs were extracted from 8.0 g soil subsamples and separated by solid-phase extraction. Methyl sulfate was used in the esterification procedure to transform the purified fatty acids into esterified molecules. Gas chromatography-mass spectrometry (Agilent 6850, Santa Clara, CA, USA) was used to examine the esterified products. The fatty acids were separated, and their mass spectra were found using GC-MS analysis. The amount of various kinds of microbial fatty acids was determined based on the peak height of the fatty acid peaks (Appendix A).

Genomic DNA was extracted from soil samples using the E.Z.N.A.^®^ Soil DNA Kit. The extracted DNA samples were kept at −20 °C. The V4 hypervariable region of the 16S rRNA gene was amplified using the universal primers 806 R (5′-GGACTACNNGGGTATCTAAT-3′) and 515F (5′-GTGCCAGCMGCCGCGGTAA-3′). The fungal ITS regions were amplified using the universal primers ITS3 (5′-GCATCGATGAAGAACGCAGC-3′) and ITS4 (5′-TCCTCCGCTTATTGATATGC-3′) [25]. The 16S rRNA gene region was selected as the focus for archaeal diversity and makeup analysis. The primers Arch349F (5′-GYGCASCAGKCCMGCCGCGGTAA-3′) and Arch806R (5′-GGACTACVSGGGTATCTAAT-3′) were used for the PCR amplification of the V3–V4 hypervariable regions of archaeal DNA. The primers utilized for the DNA of AMF were ITS1F and ITS4 (5′-TCCTCCGCTTATTGATATGC-3′) to amplify the ITS region.

PCR amplification was conducted using a Mastercycler ladder from Eppendorf, Germany. The generation of libraries from the PCR products using corresponding index tags allowed for the parallel sequencing of multiple samples. The library samples were sent to the USA-based Illumina, Inc.’s Miseq for sequencing. Following sequencing, low-quality sequences and primer sequences were discarded as part of the quality control processes for the raw data. The Uparse method was employed to perform operational taxonomic unit (OTU) clustering, which assembles related sequences into OTUs based on predetermined similarity levels. Samples were subsampled twice (dilution) to limit the effect of sequencing depth on inter-sample variance. The OTU sequences were compared to Silva 138 and Unite v8.2 databases using BLAST in order to annotate the species of the OTUs, which was applied to categorize all sequences into different genus taxonomic groups. The number of OTUs was evaluated in each subsample as microbial diversity [26]. Based on the OTUs and their abundance results obtained using QIIME (v1.8.0) software, the alpha diversity index was calculated. The number of OTUs in the soil samples served as an indicator of soil microbial diversity.

### 2.5. Statistical Analyses

To assess the simultaneous provision of multiple ecosystem functions in the temperate grassland, an EMF index was calculated by the averaging approach [27]. In this study, the EMF indexes obtained using the single-function and multiple-threshold approaches were comparable to those assessed by the averaging approach. Therefore, we used the average multifunctionality index as the EMF index. Each function (soil N mineralization, soil phosphatase activity, AMF infection rate, microbial biomass, plant biomass, and soil nutrients) was first transformed into Z-scores. The transformed values were then averaged to obtain the EMF index for each site. We assessed potential trade-off effects among multiple ecosystem functions by calculating Pearson’s correlation coefficients between each pair of single ecosystem functions. Among the 15 combinations, we found 15 significant positive correlations, and none presented a significant negative correlation, indicating no trade-off effects among them. The EMF index characterizes ecosystem multifunctionality.

To analyze the relationship between the EMF index and plant and microbial predictors, linear regression fitting methods were employed. The Shapiro-Wilk normality test was used to check for the normality of the data. Statistical analysis was conducted to compare means using Tukey’s test with a significance level of *p* < 0.05. Pearson correlation coefficients were calculated to assess the relationships between the measured variables. The “Hmisc 4.2.3” software package was used for calculating correlation coefficients, and the “ggplot2 3.4.0” software package was used for data visualization [28].

A random forest model was built using the “randomForest 4.7-1.1” software package [29]. Multiple decision tree models were constructed, each based on a random subset of features, to reduce variance and the risk of overfitting [30]. These models were then combined to make predictions. Structural equation modeling (SEM) was employed to identify the direct and indirect impacts of soil properties, plant diversity, and microbial diversity on EMF. An a priori model, representing a reasonable assumption of possible causality, was conducted for the SEM analysis (Appendix A). IBM SPSS Amos 21 was used to fit the SEM model, with the significance level set at *p* < 0.05, except for the chi-square test, where *p* > 0.05 indicated a good fit for the SEM model.

## 3. Results

### 3.1. Effects of Desertification on Biodiversity and EMF

Plant, soil bacterial, fungal, and archaeal diversity gradually decreased with increasing desertification intensity (*p* < 0.01), while arbuscular mycorrhizal fungal (AMF) diversity was not sensitive to desertification processes (Appendix A; Figure 1). Compared with the potential desertification (PD) stage, plant diversity decreased by 12.6%, 27.8%, 45.7%, 56.5%, and 55.1%; bacterial diversity decreased by 3.4%, 7.4%, 12.2%, 11.9%, and 19.5%; fungal diversity decreased by 9.7%, 14.1%, 24.0%, 24.7%, and 45.1%; and archaeal diversity decreased by 18.1%, 35.0%, 28.4%, 45.5%, and 49.0% under the light desertification (LD), moderate desertification (MD), heavy desertification (HD), severe desertification (SD), and very severe desertification (VSD) stages, respectively.

Similar to the biodiversity, the EMF also showed a similar tendency (Figure 1). As desertification intensified, EMF decreased by 46.3%, 68.8%, 114.6%, 161.7%, and 198.7% under the LD, MD, HD, SD, and VSD stages, respectively. Among the 15 ecosystem functions, soil total C, soil nutrients (N, Ca, K, Fe, Na, Mg, Cu, Mn, and P), NMR, PMA, PLFA, AMF infection rate, and plant biomass all presented to gradually decline as the desertification intensified (Figure 2).

### 3.2. Relationships between Predictors and EMF during Desertification

Linear regression showed that soil bacterial diversity (*r*^2^ = 0.802, *p* < 0.01), fungal diversity (*r*^2^ = 0.850, *p* < 0.01), and archaeal diversity (*r*^2^ = 0.342, *p* < 0.01) were all positively associated with EMF during desertification processes, except for AMF diversity (Figure 3). Similar to plant diversity, microbial diversity was also positively correlated with EMF across different desertification stages (*r*^2^ = 0.837, *p* < 0.01).

For the soil property predictors, the regression analyses also showed that EMF was positively correlated with soil water content (*r*^2^ = 0.863, *p* < 0.01) and soil clay content (*r*^2^ = 0.811, *p* < 0.01) but was negatively associated with soil pH (*r*^2^ = 0.683, *p* < 0.01; Figure 3). Furthermore, plant diversity significantly and positively correlated with soil bacterial diversity (*r*^2^ = 0.654, *p* < 0.01), fungal diversity (*r*^2^ = 0.654, *p* < 0.01), and archaeal diversity (*r*^2^ = 0.285, *p* < 0.01).

### 3.3. Relative Importance of Biodiversity and Abiotic Factors on EMF during Desertification

In order to determine the significance of biodiversity and soil property predictors of EMF under grassland desertification processes (plant diversity, bacterial diversity, fungal diversity, archaeal diversity, arbuscular mycorrhizal fungi (AMF) diversity, soil water content, soil clay content, and soil pH), we first employed a random forest model. The soil water content, plant diversity, fungal diversity, soil clay content, bacterial diversity, and AMF diversity were the most significant predictors in controlling EMF during desertification (*p* < 0.01), followed by soil pH (*p* < 0.05), according to the random forest model (*r*^2^ = 0.98, *p* < 0.001). In contrast, there was no significant link between soil archaeal diversity and EMF (Figure 4).

The relationships between biodiversity (plant, bacterial, fungal, archaeal, and AMF diversity) and EMF were then examined using structural equation modeling (SEM; see the a priori model in Appendix A) to see if they remained constant when soil properties were also taken into account. Under grassland desertification processes, the SEM explained 97% of the variance in the EMF (*Chi-squire*/*df* = 0.946, *p* = 0.483, *AIC* = 110.000; Figure 5). We discovered that plant diversity, soil microbial diversity, soil water content, and soil clay content all directly and indirectly influenced EMF. Furthermore, only soil pH had no significant effect on EMF when desertification processes occurred. Although AMF diversity has a direct impact on EMF, it does not appear to be impacted by the processes of desertification.

## 4. Discussion

Our findings established that increasing desertification significantly declined plant diversity, soil microbial diversity, and ecosystem multifunctionality (EMF) (Figure 1 and Figure 3). Furthermore, we observed that both plant and soil microbial diversity were critical for regulating the response of EMF to desertification processes in temperate grasslands.

### 4.1. Impact of Desertification on Plant Diversity

Consistent with our first hypothesis, we observed a gradual reduction in plant diversity with increasing desertification intensity (Figure 1). This finding is consistent with recent studies conducted in temperate grasslands [31]. Several explanations underscore this observation. Firstly, grassland desertification leads to the decline of soil moisture and soil nutrient storage, which contributes to the loss of specific plant species that are unable to adapt to water and nutrient limitations in temperate grassland [32,33]. Secondly, desertification processes may be accompanied by the cumulative effects of wind erosion in semiarid regions. The low plant coverage in the desertified grasslands is exacerbated by the impact of wind erosion on the soil clay content [34,35] and may inhibit the growth of specific plant species [36]. Consequently, once the plant diversity is lost, the grassland soils will continue to degrade, resulting in soil C and nutrient instability and accelerating desertification processes. These alterations adversely impact ecological construction as well as economic and social development [37].

### 4.2. Impact of Desertification on Soil Microbial Diversity

Similar to the changes in plant diversity, our results also demonstrated that soil bacterial, fungal, and archaeal diversity gradually decreased with increasing desertification intensity (Figure 1), consistent with previous studies [33]. There are several explanations for these observations. Firstly, low soil moisture and high soil sand content during desertification may inhibit the growth of anaerobic bacteria by increasing the oxygen content [38]. Secondly, plant diversity can predict soil microbial taxonomic diversity in natural and managed ecosystems [2,39]. Therefore, the significant decreases in soil bacterial, fungal, and archaeal diversity are likely because of the positive plant-microbe relationships during desertification in our study (Figure 1). Furthermore, soil C storage is considered a dominant determinant of microbial diversity [7]. As plant biomass decreased during desertification, there was a subsequent reduction in litter input and soil C storage, which may result in a significant loss in microbial diversity [5]. In contrast, AMF relies on symbiotic mechanisms and obtains plant photosynthetic C from host plants in exchange for other nutrients [40]. This unique relationship may contribute to their resistance and reduced variation compared to bacterial and fungal diversity (Figure 3).

### 4.3. Impact of Desertification on EMF

Consistent with our second hypothesis, we identified a significant decline in EMF with increasing desertification intensity (Figure 1). This could be attributed to substantial reductions in multiple ecosystem functions, including soil nutrients, plant biomass, and C storage during desertification (Figure 2). This finding was consistent with studies conducted in arid and semiarid grasslands, which have reported that desertification significantly reduced many ecosystem functions, including productivity and nutrient storage [16,31]. Firstly, the decrease in soil clay content in the desertified grasslands leads to a reduction in soil nutrients as soil organic matter is combined with soil fine fractions [41]. Secondly, desertification-induced loss of water and nutrients ultimately leads to limited plant productivity [42]. Thirdly, significant decreases in productivity could limit soil nutrient inputs from litter. In addition, increases in soil porosity can increase water infiltration, amplifying the risk of soil organic C loss via water erosion [43]. In general, EMF is sensitive and vulnerable to increasing desertification intensity within the temperate grassland ecosystem [2].

### 4.4. Biodiversity Regulates EMF during Desertification

Consistent with our third hypothesis, we identified that plant and microbial diversity were positively correlated with EMF during desertification (Figure 3). Our SEM indicated that plant, bacterial, and fungal diversity played clear roles in mediating EMF during desertification (Figure 3, Figure 4 and Figure 5). The result was in agreement with previous studies conducted in drylands and grasslands, which demonstrated that plant richness is closely tied to multiple ecosystem functions, including net primary production [44,45], C sequestration [46], nutrient cycling [47], and soil respiration [17,38,39,40]. Therefore, the reduction in plant diversity with increasing desertification intensity could limit EMF through biodiversity-regulated effects in temperate grassland [48].

Similar to plant diversity, soil bacterial and fungal diversity also played crucial roles in directly regulating EMF during desertification (Figure 3). This is consistent with previous studies, which reported a positive correlation between soil microbial diversity and EMF due to niche differentiation among diverse species having considerable potential to simultaneously stimulate multiple functions [5,49]. In addition, the positive impact of fungal diversity on EMF throughout desertification was slightly higher than that of bacterial diversity (Figure 4 and Figure 5). This is because soil fungi are more tolerant to desiccation than bacteria, and fungal diversity may, therefore, have a larger impact on EMF [7,50]. This finding indicates that soil microbial diversity is primary to linking the above- and below-ground functions in desertified grasslands.

To contribute to ammonia oxidation, soil archaea, which are important participants in the global C and N cycles, have congeners and subunits of the bacterial ammonia monooxygenases [51]. Therefore, the reduction in archaeal diversity may lead to a reduction in EMF in the temperate grassland (Figure 3). However, our random forest model indicates that archaeal diversity is not a significant predictor during desertification processes. In contrast, soil bacteria and fungi have broader taxonomic groups than archaea and AMF, with relatively diverse association traits and functions. The similarities in taxonomic composition, morphological structure, and biochemical metabolism between archaea and bacteria likely contribute to a co-linear relationship (Figure 4).

Our study is consistent with prior studies suggesting that abiotic predictors also play essential roles in grassland desertification (Appendix A). For the soil property predictors, our random forest and structural equation models further showed that soil water content was the most significant factor impacting EMF during desertification (Figure 4 and Figure 5). Soil water availability is a crucial abiotic predictor for ecosystem functions in degraded grasslands [52]. A recent study demonstrated that increased precipitation enhances biodiversity in desert grasslands, which is positively correlated with EMF [4]. Additionally, grassland desertification can limit soil clay content (Appendix A). Low soil clay content may have adverse impacts on the physical and chemical properties of soil [27].

In contrast, plant diversity appears to have a greater influence on EMF than bacterial and fungal diversity throughout the desertification processes (Figure 4 and Figure 5). This finding could be explained by the comparatively delayed response of soil microbial communities to regulating soil functions [2]. Compared to plants, soil microbes have increased resistance and resilience to climate change and disturbance because they can utilize both fresh and old organic matter under harsh conditions [53]. Generally, both plant and microbial diversity play a cooperative and integral role in regulating EMF.

### 4.5. Limitations and Future Work

We have addressed several limitations in our study that should be considered in future work. First, studies on the biodiversity-multifunctionality relationship during desertification have primarily focused on the topsoil layers, ignoring possible dynamics in the deep soil layers. Second, although the EMF index is useful for ecological research, it is necessary to recognize the uncertainty in assessing multifunctionality accurately. Therefore, it is crucial to consider any biases that might be created by the measurement methods and ecosystem functions chosen for EMF assessment. Future studies should develop a standardized technique (e.g., cluster analysis) and focus on independent aspects of ecosystem functions (e.g., nutrient storage and decomposition).

## 5. Conclusions

Our findings showed that the EMF, plant, soil bacterial, fungal, and archaeal diversity gradually decreased with the increasing desertification intensity in the temperate grassland. However, arbuscular mycorrhizal fungal diversity was not affected by desertification. We found that plant and soil microbial diversity co-regulated the responses of EMF to desertification processes. This study provides innovative insights into teasing apart the influences of plant and microbial diversity in mediating the impact of desertification on EMF in temperate grassland. This highlights that any loss in plant and microbial diversity resulting from grassland desertification would undermine EMF. This finding has important implications for degraded ecosystem restoration and sustainable management in grassland ecosystems.

## Figures and Tables

**Figure 1 plants-12-03743-f001:**
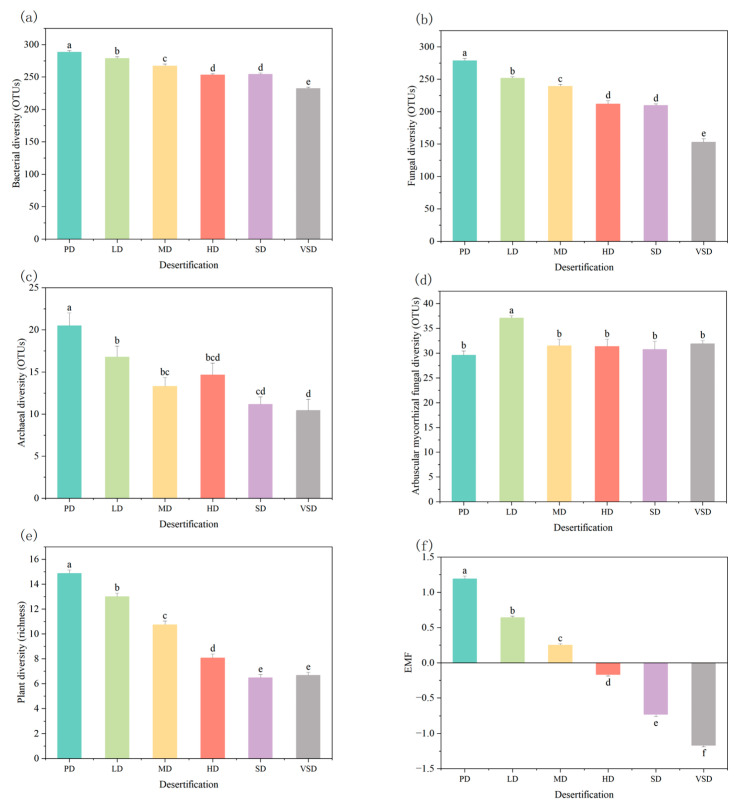
Soil bacterial diversity ((**a**) OTUs), fungal diversity ((**b**) OTUs), archaeal diversity ((**c**) OTUs), arbuscular mycorrhizal fungal diversity ((**d**) OTUs), plant diversity ((**e**) richness), and ecosystem multifunctionality index (EMF, (**f**)) under different desertification stages. Significant differences (*p* < 0.05) among different desertification stages are indicated by different lowercase letters. PD, potential desertification; LD, light desertification; MD, moderate desertification; HD, heavy desertification; SD, severe desertification; and VSD, very severe desertification.

**Figure 2 plants-12-03743-f002:**
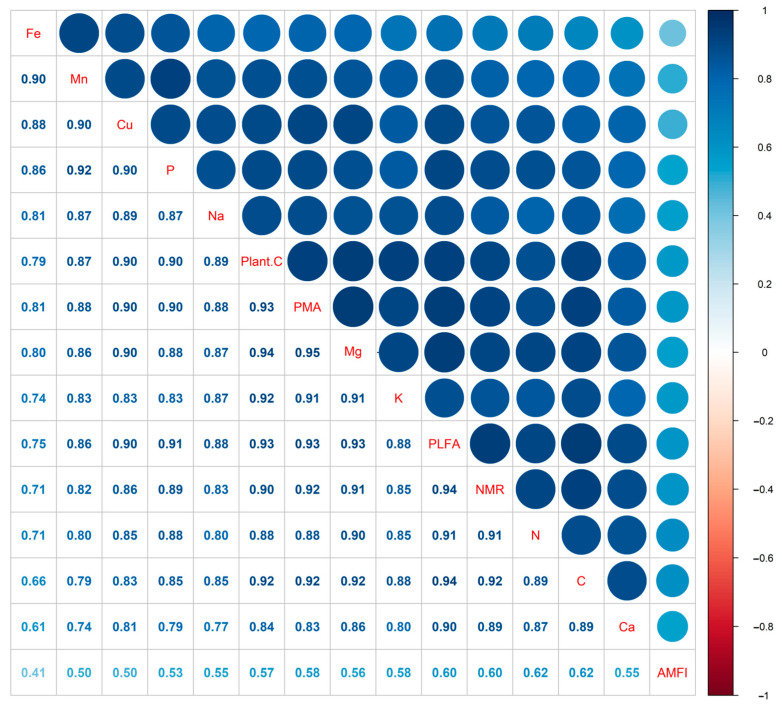
Correlation heat maps show relationships between ecosystem functions (soil total C, soil nutrients (N, Ca, K, Fe, Na, Mg, Cu, Mn, and P), soil nitrogen mineralization (NMR), phosphatase activity (PMA), phospholipid fatty acid (PLFA), arbuscular mycorrhizal fungal infection rate (AMFI), and plant biomass) during desertification processes.

**Figure 3 plants-12-03743-f003:**
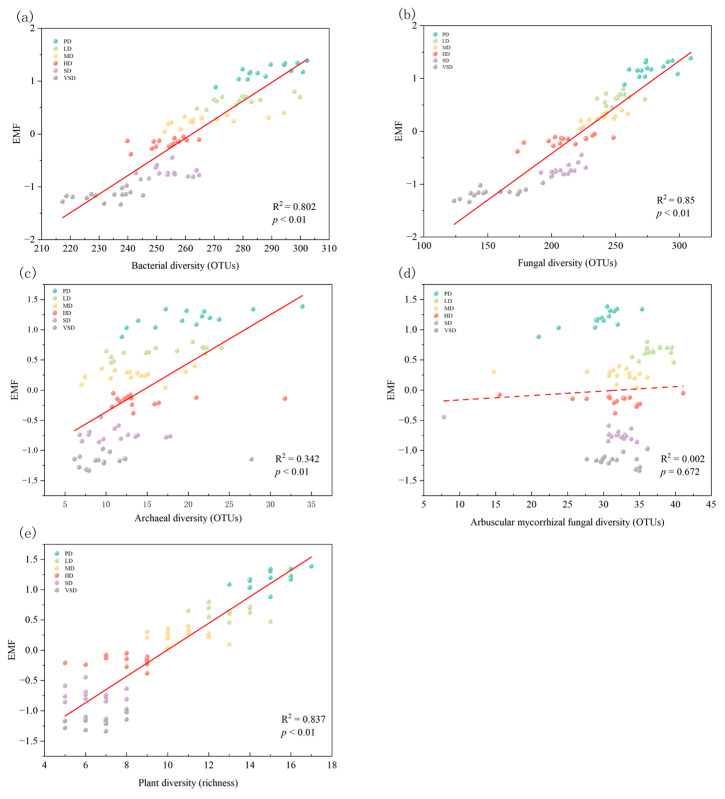
Relationships between ecosystem multifunctionality (EMF) and bacterial diversity ((**a**) OTUs), fungal diversity ((**b**) OTUs), archaeal diversity ((**c**) OTUs), arbuscular mycorrhizal diversity ((**d**) OTUs), and plant diversity ((**e**) richness). The solid lines represent the fitted ordinary least squares.

**Figure 4 plants-12-03743-f004:**
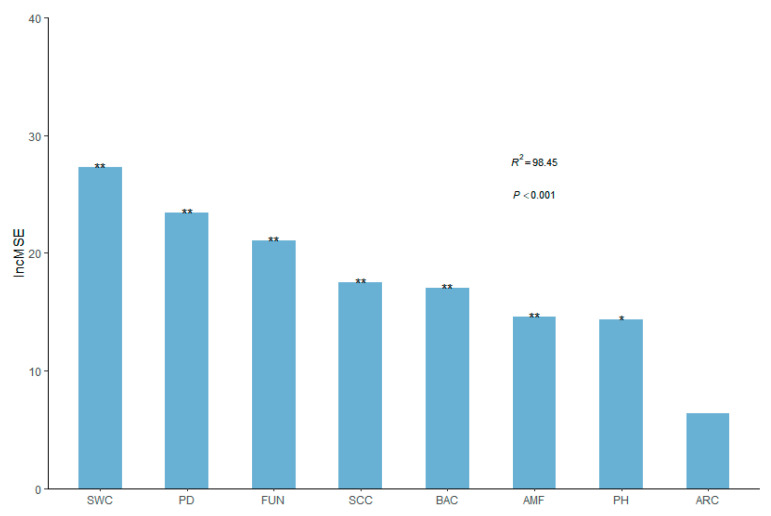
Main predictors of ecosystem multifunctionality (EMF). The figure shows the random forest mean predictor importance (% of increase in MSE) of biodiversity and abiotic factors on EMF for temperate grassland. The significance levels of each predictor are as follows: * *p* < 0.05 and ** *p* < 0.01. SWC, soil water content; PD, plant diversity; FUN, fungal diversity; SCC, soil clay content; BAC, bacterial diversity; AMF, arbuscular mycorrhizal fungal diversity; ARC, archaeal diversity.

**Figure 5 plants-12-03743-f005:**
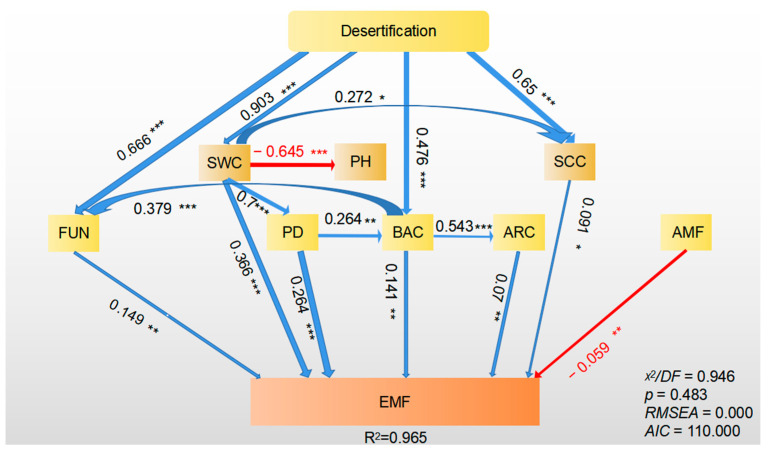
The structural equation model is shown for the direct and indirect effects of soil water content, soil clay content, soil pH, bacterial diversity, fungal diversity, archaeal diversity, arbuscular mycorrhizal fungal diversity, and plant diversity on ecosystem multifunctionality. Numbers adjacent to arrows are indicative of the effect size (bootstrap *p* value) of the relationship. Significant positive and negative effects are shown with blue and red arrows, respectively. The width of arrows is proportional to the strength of path coefficients. R^2^ denotes the proportion of variance explained. EMF, ecosystem multifunctionality; SWC, soil water content; SCC, soil clay content; PD, plant diversity; FUN, fungal diversity; BAC, bacterial diversity; AMF, arbuscular mycorrhizal fungal diversity; PH, soil pH; ARC, archaeal diversity. * represents *p* < 0.05, ** represents *p* < 0.01, *** represents *p* < 0.001.

## Data Availability

All data are included in the manuscript and Supporting Information section.

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
