# Peer review of "Plant and Soil Microbial Diversity Co-Regulate Ecosystem Multifunctionality during Desertification in a Temperate Grassland"

_plants, 2023, doi:10.3390/plants12213743_

Round 1

Reviewer 1 Report (Previous Reviewer 3)

Comments and Suggestions for Authors

ere are several units of diversity but they failed to clarify what they were referring to. Secondly, I still can't follow the introduction and discussion as it is communication is poor. On the methods, the authors used UPARSE for OTU clustering which is not consistent with recent methods like DADA2 to obtain ASVs. The authors mentioned in the methods that they did taxonomic assignment but none of that data nor anything related to ordination was presented.

Comments on the Quality of English Language

The quality of the English language is very and requires extensive revision

Author Response

Dear Reviewer,

Your scientific comments have been invaluable for improving our manuscript, and each point has been fully addressed. The detailed point-by-point responses are listed as below. We also used the Editing Service to help us polish our MS. We appreciate your consideration of our manuscript.

Sincerely,

Linna Ma, Yeming Zhang

Responses: 

  1. Thank you for pointing this out. We used the richness and OTUs to assess plant and microbial diversity. In the old MS version, we did not clearly explain the units of biodiversity due to our misunderstanding. In the revised MS, we added the units of richness and OTUs in the M&M, Figures, Results and Figure Legends. Please see lines 131, 182-185, Figure 1, 3.
  2. Thank you for your helpful suggestion. We have partly revised the Introduction and Discussion in order to enhance the coherence and informativeness between the Introduction and Discussion. Please see lines 55-60, 63-69, 73-81, 278-285, 313-320, and 363-364.
  3. Thank you for addressing this important point. ASV is a relatively new method based on error correction and detection of precise sequence variants during gene sequencing. It is a great method and we will use it in our future studies.
  4. The metadata of soil microbial communities will be provided in the Dryad Digital Repository after the manuscript is accepted. Thank you for your support and understanding.
  5. We used the Editing Service to help us polish the MS.

Reviewer 2 Report (New Reviewer)

Comments and Suggestions for Authors

The research described in the manuscript concerns the response of microorganisms and plants to grassland desertification and changes in the electromagnetic field under desertification conditions.
The work is a scientific article. It has the correct structure. It is consistent with the profile of the magazine. However, methodological information needs to be supplemented, because only on the basis of a more detailed description can the substantive value of the results, their interpretation and conclusions be correctly assessed.
Detailed comments
Chapter 2 "Material and methods" lacks information on how to measure the electromagnetic field.
In chapter 2.5. "Statistical analyses" should describe in more detail how to calculate the EMF index. An interpretation of the value of this indicator should also be presented.
Chapter 2.4. should be supplemented with information about which region of 16S rRNA was examined.
Information about the database used to calculate the Shannon diversity index also requires supplementation. The taxonomic level of the microorganisms should be provided.
In Figure 4, instead of **, I propose to include homogeneous groups marked with letters.
Table S2 and S3 I suggest entering homogeneous groups.
Table S7 – zeros are missing before the dot of the entered coefficients.

Author Response

Reviewer: 2

COMMENTS FOR THE AUTHOR

The research described in the manuscript concerns the response of microorganisms and plants to grassland desertification and changes in the electromagnetic field under desertification conditions.
The work is a scientific article. It has the correct structure. It is consistent with the profile of the magazine. However, methodological information needs to be supplemented, because only on the basis of a more detailed description can the substantive value of the results, their interpretation and conclusions be correctly assessed.

Dear Reviewer,

The scientific comments have been invaluable for improving our manuscript, and each point has been fully addressed in the revised MS. We also used the Editing Service to help us polish the MS. The detailed point-by-point responses are listed as below.

We appreciate your consideration of our manuscript.

Sincerely,

Linna Ma, Yeming Zhang

 Detailed comments
Chapter 2 "Material and methods" lacks information on how to measure the electromagnetic field.

Response: Thank you for pointing this out. Sorry, I couldn’t understand the meaning of “electromagnetic field” in this question, does it mean EMF? EMF index was calculated by the averaging approach. In contrast, the EMF also obtained using the single-function and multiple-threshold approaches which comparable to those assessed by the averaging approach. Therefore, this study used the averaged multifunctionality index as the EMF index. Please see the detailed information in the revised manuscript (lines188-191).

In chapter 2.5. "Statistical analyses" should describe in more detail how to calculate the EMF index. An interpretation of the value of this indicator should also be presented.

Response: Thank you for your helpful suggestion. In this study, each ecosystem function (soil N mineralization, soil phosphatase activity, AMF infection rate, microbial biomass, plant biomass, and soil nutrients) was firstly transformed into Z-scores. The transformed values were then averaged to obtain the EMF index for each site. We assessed potential trade-off effects among multiple ecosystem functions by calculating Pearson's correlation coefficients between each pair of single ecosystem functions. Among the 15 combinations, we found no significant trade-off effects among them. The details were added in the section of Statistical analyses. Please see lines 188-191 and 194-199.

Chapter 2.4. should be supplemented with information about which region of 16S rRNA was examined.

Response: Thank you for pointing this out. PCR amplification of the V4 hypervariable regions of bacteria and the V3-V4 hypervariable regions of archaea was performed using 16S rRNA. Please see the detailed information in the revised manuscript (line 158 and line 167).

Information about the database used to calculate the Shannon diversity index also requires supplementation. The taxonomic level of the microorganisms should be provided.

Response: Thank you for your helpful suggestion. The OTU sequences were compared to Silva 138 and unite v8.2 databases using BLAST in order to annotate the species of the OTUs, which was applied to categorize all sequences into different genus taxonomic groups. In the old MS version, our expression is vague, and thus we rewrote this part. Actually, the number of OTUs in the soil samples served as an indicator of soil microbial diversity, and the plant richness served as an indicator of plant diversity in this study. Please see them in the revised manuscript (lines 179-181). In addition, the metadata of soil microbial communities will be provided in the Dryad Digital Repository after the manuscript is accepted. Thank you for your support and understanding.

In Figure 4, instead of **, I propose to include homogeneous groups marked with letters.

Response: Thank you for pointing this out. Actually, Random Forest model calculates the degree of importance of each feature. Therefore, we hope to use the “**” to show the significant predictors rather than letters. Thank you for your understanding.

Some recent studies have also used similar method to show the results of Random Forest model, including Wang et al. (2023, Global Ecology & Biogeography 2022, 31, 886–900), Zhang et al. (Applied Soil Ecology 2023, 189, 104953), and Delgado-Baquerizo et al. (2016, Nature Communication, 7, 10541).

  1. Wang, C.; Ma, L.; Zuo, X.; Ye, X.; Wang, R.; Huang, Z.; Liu, G.; Cornelissen, J.H.C. Plant Diversity Has Stronger Linkage with Soil Fungal Diversity than with Bacterial Diversity across Grasslands of Northern China. Global Ecology and Biogeography2022, 31, 886–900, doi:10.1111/geb.13462.
  2. Zhang, Y.; Li, X.; Zhang, J.; Hua, J.; Li, J.; Liu, D.; Bhople, P.; Ruan, H.; Yang, N. Desertification Induced Changes in Soil Bacterial and Fungal Diversity and Community Structure in a Dry-Hot Valley Forest. Applied Soil Ecology2023, 189, 104953, doi:10.1016/j.apsoil.2023.104953.
  3. Delgado-Baquerizo, M.; Maestre, F.T.; Reich, P.B.; Jeffries, T.C.; Gaitan, J.J.; Encinar, D.; Berdugo, M.; Campbell, C.D.; Singh, B.K. Microbial Diversity Drives Multifunctionality in Terrestrial Ecosystems. Nat Commun2016, 7, 10541, doi:10.1038/ncomms10541.

Table S2 and S3 I suggest entering homogeneous groups.

Response: Thank you for pointing this out. We have revised Table S2 and S3 based on your suggestion. Significant differences among desertification intensity are indicated by different lower-case letters in the same column (p < 0.05). For more details, please see Table S2 and S3 in Supporting information.

Table S7 – zeros are missing before the dot of the entered coefficients.

Response: The table was exported directly through our statistical analysis software and this detail was overlooked, thank you for your suggestion, it has been corrected here. Please see Table S7.

Reviewer 3 Report (Previous Reviewer 1)

Comments and Suggestions for Authors

In general, I am satisfied the corrections and responses given by the authors. I recommend to publish this manuscript.

Comments on the Quality of English Language

The authors should pay more attention to the English expression with appropriate tense, i.e. the present and past tense. There are spaces for English improvement.

Author Response

Dear Reviewer,

We appreciate your encouragement for this revised manuscript. Your scientific comments have been invaluable for improving our manuscript.  We used the Editing Service to help us polish the MS. 

We appreciate your consideration of our manuscript.

Sincerely,

Yeming Zhang, Linna Ma

This manuscript is a resubmission of an earlier submission. The following is a list of the peer review reports and author responses from that submission.

Round 1

Reviewer 1 Report

Comments and Suggestions for Authors

The manuscript titled "Soil microbial and plant diversity co-regulate ecosystem multifunctionality during desertification in a temperate grassland" tried to address how biodiversity, including plant and microbial diversity, regulated the the impacts of desertification processes on ecosystem multifunctionality. The authors used correlation analysis, random forest and structural equation model to identify the main factors affecting multifunctions in a temperate grassland. Although the conclusion "microbial and plant diversity co-regulated the responses of ecosystem multifunctionality to desertification" is interesting, but the authors just disclosed a phenomenon, but did not provide the mechanisms or discussions to explain why this phenomenon occurs. I did not find the novel points of this manuscript. My major concerns are as follows:
    First,you used 15 indicators of so-called ecosystem functions, you should clarify what kind of ecosystem function are related. In the 15 indicators, I find that 11 indicators, indicating soil available N and nutrients are related to soil nutrient, had excessively high proportion in the “functions”, which may take overwhelming effects on soil nutrient. You should not mix everything, what you measured, to be multifunctions, like a pot of stew! Other functions cannot rival the weight of nutrient in this study. So I feel your identification of multifunctions is farfetched.
    Second, how are the successive stages of desertification in the study area are classified? You should have a general information of the community composition and soil conditions for the series of succession. And how are plant diversity and microbial diversity are defined and calculated?

Third, some of contents in the Discussion section seems to be speculative. For example, "The loss of plant diversity during desertification may be attributed to certain herbaceous species and vulnerable forb species that are unable to adapt to soil water and nutrient limitations, leading to a loss of their ability to survive", you have not data to support your argument. The discussion did not unravel the mechanism of diversity-multifunctionality but a superficial correlation analysis.
    Fourth, why plant diversity is more important than microbial diversify in determining multifunctions? and why AMF has negative impact on multifunctions? You should have a clear explanation of the causes.

Overall, sorry, I cannot provide a positive recommendation for this manuscript.

Specific Comments:
    1. line 79-84, for the hypotheses, in particular the third one, you should have reasonable inference and propose the reasonable hypotheses.

2. Line 209-211, “Compared with the biodiversity, the EMF also showed similar tendency (Figure 1). As desertification intensified, EMF decreased by 46.3%, 68,8%, 114.6%, 161.7%, 198.7% under PD, LD, MD, HD, SD and VSD stages, respectively.” what is the reference for the comparison? You list five values for six stages.

Comments on the Quality of English Language

The manuscript titled "Soil microbial and plant diversity co-regulate ecosystem multifunctionality during desertification in a temperate grassland" tried to address how biodiversity, including plant and microbial diversity, regulated the the impacts of desertification processes on ecosystem multifunctionality. The authors used correlation analysis, random forest and structural equation model to identify the main factors affecting multifunctions in a temperate grassland. Although the conclusion "microbial and plant diversity co-regulated the responses of ecosystem multifunctionality to desertification" is interesting, but the authors just disclosed a phenomenon, but did not provide the mechanisms or discussions to explain why this phenomenon occurs. I did not find the novel points of this manuscript. My major concerns are as follows:
    First,you used 15 indicators of so-called ecosystem functions, you should clarify what kind of ecosystem function are related. In the 15 indicators, I find that 11 indicators, indicating soil available N and nutrients are related to soil nutrient, had excessively high proportion in the “functions”, which may take overwhelming effects on soil nutrient. You should not mix everything, what you measured, to be multifunctions, like a pot of stew! Other functions cannot rival the weight of nutrient in this study. So I feel your identification of multifunctions is farfetched.
    Second, how are the successive stages of desertification in the study area are classified? You should have a general information of the community composition and soil conditions for the series of succession. And how are plant diversity and microbial diversity are defined and calculated?

Third, some of contents in the Discussion section seems to be speculative. For example, "The loss of plant diversity during desertification may be attributed to certain herbaceous species and vulnerable forb species that are unable to adapt to soil water and nutrient limitations, leading to a loss of their ability to survive", you have not data to support your argument. The discussion did not unravel the mechanism of diversity-multifunctionality but a superficial correlation analysis.
    Fourth, why plant diversity is more important than microbial diversify in determining multifunctions? and why AMF has negative impact on multifunctions? You should have a clear explanation of the causes.

Overall, sorry, I cannot provide a positive recommendation for this manuscript.

Specific Comments:
    1. line 79-84, for the hypotheses, in particular the third one, you should have reasonable inference and propose the reasonable hypotheses.

2. Line 209-211, “Compared with the biodiversity, the EMF also showed similar tendency (Figure 1). As desertification intensified, EMF decreased by 46.3%, 68,8%, 114.6%, 161.7%, 198.7% under PD, LD, MD, HD, SD and VSD stages, respectively.” what is the reference for the comparison? You list five values for six stages.

Reviewer 2 Report

Comments and Suggestions for Authors

The manuscript nt" Soil microbial and plant diversity co-regulate ecosystem multifunctionality during desertification in a temperate grassland"  is very interesting and provide new findings. The subject is within the scope of the journal and the aim of the study is relevant. The introduction, material and methods, results, discussion section and conclusions is well written. English language and references do not raise objections. I recommend publishing the manuscript in a journal. 

Reviewer 3 Report

Comments and Suggestions for Authors

The introduction is not coherent. There needs to be a flow of ideas. The discussion needs to be rewritten as it is a rehash of the results without proper discussions.

Some specific comments

L24-24: Rephrase. Multiply drivers in driving EMF?

L33-34: By contrast, plant diversity exerting a stronger influence than bacterial and fungal diversity. This is confusing

L57-60: I don’t see the link.

L132-133: Please rephrase

L162-169: This is not consistent with current methods

L203-204: What is the unit of your diversity?

L216-218: Why do you have figure 3 before figure 2?

Comments on the Quality of English Language

Some of the sentences need to be rephrased in order to capture the meaning